# TGF-β Signaling: From Tissue Fibrosis to Tumor Microenvironment

**DOI:** 10.3390/ijms22147575

**Published:** 2021-07-15

**Authors:** Jeff Yat-Fai Chung, Max Kam-Kwan Chan, Jane Siu-Fan Li, Alex Siu-Wing Chan, Philip Chiu-Tsun Tang, Kam-Tong Leung, Ka-Fai To, Hui-Yao Lan, Patrick Ming-Kuen Tang

**Affiliations:** 1Department of Anatomical and Cellular Pathology, State Key Laboratory of Translational Oncology, The Chinese University of Hong Kong, Hong Kong, China; jeffchung@link.cuhk.edu.hk (J.Y.-F.C.); maxchankamkwan@link.cuhk.edu.hk (M.K.-K.C.); jane.li@link.cuhk.edu.hk (J.S.-F.L.); philtang@link.cuhk.edu.hk (P.C.-T.T.); kfto@cuhk.edu.hk (K.-F.T.); 2Department of Applied Social Sciences, The Hong Kong Polytechnic University, Hong Kong, China; alexsw.chan@connect.polyu.hk; 3Department of Paediatrics, Prince of Wales Hospital, The Chinese University of Hong Kong, Hong Kong, China; ktleung@cuhk.edu.hk; 4Department of Medicine & Therapeutics, Li Ka Shing Institute of Health Sciences, Lui Che Woo Institute of Innovative Medicine, The Chinese University of Hong Kong, Hong Kong, China; hylan@cuhk.edu.hk

**Keywords:** TGF-β, tumor microenvironment, fibrosis

## Abstract

Transforming growth factor-β (TGF-β) signaling triggers diverse biological actions in inflammatory diseases. In tissue fibrosis, it acts as a key pathogenic regulator for promoting immunoregulation via controlling the activation, proliferation, and apoptosis of immunocytes. In cancer, it plays a critical role in tumor microenvironment (TME) for accelerating invasion, metastasis, angiogenesis, and immunosuppression. Increasing evidence suggest a pleiotropic nature of TGF-β signaling as a critical pathway for generating fibrotic TME, which contains numerous cancer-associated fibroblasts (CAFs), extracellular matrix proteins, and remodeling enzymes. Its pathogenic roles and working mechanisms in tumorigenesis are still largely unclear. Importantly, recent studies successfully demonstrated the clinical implications of fibrotic TME in cancer. This review systematically summarized the latest updates and discoveries of TGF-β signaling in the fibrotic TME.

## 1. Introduction

Transforming growth factor-beta (TGF-β) was discovered as a versatile cytokine with both pathological and physiological functions. The main isoforms of the TGF-β family, TGF-β1, TGF-β2, and TGF-β3, show different biological activities [1,2]. Interestingly, only the promoter region of TGF-β1 can be activated directly by different trans-activating proteins such as reactive oxygen species (ROS), plasmin, and acid due to its multiple regulatory sites [3], highlighting its pleiotropic nature in fibrogenesis, carcinogenesis, immune modulation, cell proliferation, and differentiation [4,5].

The molecular pathways of TGF-β signaling have been studied extensively in various biological processes, including proliferation, differentiation, and apoptosis [6]. TGF-β signaling causes different downstream actions in a context dependent manner, especially in cancer. It plays a dual role, being both a tumor suppressor in pre-malignant cells and a tumor promoter in cancer cells, respectively [7]. Cancer cells are able to inactivate the tumor suppressive components of the TGF-β/Smad signaling through acquired mutations, while the tumor suppressive effects can exert selective pressure on the pre-malignant cells [8]. Indeed, cancer cells can also use the pleiotropic nature of TGF-β signaling and its downstream mediators to escape from the anti-tumor immunity by creating an immunosuppressive tumor microenvironment (TME) [8,9]. The TGF-β/Smad pathway mediates endothelial-mesenchymal transformation via SNAIL/Slug expression in tumor endothelial cells (TECs) to support sprouting angiogenesis, and accumulation of myofibroblast and CAF in TME [10].

Fibrosis is defined as an excessive deposition of fibrous connective tissue. Accumulation of extracellular matrix (ECM) components disrupts the physiological architecture and function of the normal tissues, leading to morbidity and mortality. At both the physiological and pathological levels, TGF-β is a master regulator of fibrogenesis [11]. Indeed, sole activation of TGF-β signaling is sufficient to cause fibrosis, e.g., forced overexpression of a constitutively active TGF-β receptor type I in murine skin is capable of inducing fibrosis [12]. Importantly, contribution of fibrotic signaling in the TME has recently been recognized and is suggested to be a new therapeutic target for blocking cancer progression [13]. This review summarized the pathogenic roles and underlying mechanisms of TGF-β signaling in cancer, particularly the fibrotic TME.

## 2. TGF-β Signaling

The TGF-β superfamily was discovered as a group of cytokines with shared properties in synthesis, signal transduction mechanisms, and functions [14,15]. Eventually, three isoforms, TGF-β1, TGF-β2, and TGF-β3, were eventually identified, synthesized as inactive precursor forms that need to be further activated for triggering their downstream canonical or non-canonical pathways [7]. C-terminal TGF-β homodimer, latency-associated peptide (LAP), and N-terminal signal peptide are three components of the TGF-β precursor. Cleavage of this precursor removes the signal peptide, leaving the TGF-β homodimer and LAP together, known as the small latent complex (SLC) [7,11]. Then, the SLC form a disulfide linkage to the latent TGF-β binding protein (LTBP), leading to the production of a large latent complex (LLC), which is often associated with ECM and remains inactive [7]. The LLC can be further cleaved by proteases to release the active TGF-β homodimer for activating its receptors to initiate the downstream signaling. 

### 2.1. Canonical Pathway

Conventionally, the active TGF-β binds to the TGF-β receptor type II, which recruits and phosphorylates type TGF-β receptor type I (TGF-βR1) [11]. The activated TGF-βR1 will phosphorylate the C-terminal serine residues of the receptor-associated Smads (R-Smads) Smad2 and Smad3, then, the R-Smads will dissociate from type I receptor and form a heterotrimeric complex with common Smad (Co-Smad) Smad4 at the early endosome and translocate into the nucleus [14,16]. The Smad complexes will interact with transcription factors, chromatin binding proteins and transcription co-activators and co-repressors in the nucleus and physically bind to the target genes for executing the transcriptional regulation at genomic level [8,16], resulting in the diverse effects of the TGF-β canonical pathway in different contexts. There is also another Smad member called inhibitory Smad (I-Smad), e.g., the classic example Smad7, which negatively regulates the canonical pathway by competing with Smad 2/3 for binding of type I receptor as a negative feedback for attenuating the TGF-β/Smad signaling [11], as summarized in Figure 1.

In canonical signaling cascade, the activated TGF-βR1 leads to Smad2/3 phosphorylation. The Smad2/3 complex then binds to Smad4 and translocates into the nucleus, where it induces gene transcription. The signaling can be inhibited by a negative feedback of Smad7 on the TGF-βR1.

### 2.2. Non-Canonical Pathway

Besides, there are non-canonical pathways of the TGF-β1 signaling, where extracellular signal-regulated kinase (Erk)/mitogen-activated protein kinase (MAPK) signaling pathway are involved in the downstream signal transductions. After the phosphorylation of TGF-β type I and II receptors at tyrosine residues, adaptor proteins Src homology 2 (SH2) domain-containing protein A (ShcA) and growth factor receptor-bound protein 2 (Grb2)/son of sevenless (SOS) complex is recruited [17]. The Grb2/SOS complex catalyzes the exchange of GDP to GTP for activating a small GTPase Ras, which then triggers the gene regulatory actions via Raf, MEK1/2, and Erk1/2 [17,18]. For example, Erk1/2 can phosphorylate targeted transcription factors such as Fos-related antigen 2 (Fra-2) in order to promote gene transcription [18,19]. In addition, protein kinase B (Akt) can be activated by TGF-β signaling via phosphatidylinositol-3 kinase (PI3K) for regulating translational responses though mTOR [17]. Akt can also be activated in the non-canonical pathway via TRAF6-mediated Akt lysine-63 chain ubiquitination or Smad7 phosphatase and tensin homolog (PTEN) inhibition by miR-216a/217 microRNA cluster [20,21]. Moreover, RhoA and Rho-associated protein kinase (ROCK) can also been regulated by TGF-β signaling through a Smad independent manner [17]. In addition, TGF-β mediates epithelial-mesenchymal transformation (EMT) of cancer cells by interfering with cell adhesion and epithelial gene expression, as well as by increasing the expression of mesenchymal proteins such as fibronectin, N-cadherin, vimentin, and fibronectin [22]. Activation of the Smad-dependent pathway induces the expression of SNAIL, SLUG, ZEB, and TWIST transcription factors, which act as E-cadherin repressors and mediate the dissociation of desmosomes [23]. On the other hand, activation of Smad-independent pathway promotes cytoskeletal remodelling by ERK activation and tight junction dissolution via Rho GTPase [24]. ERK strengthens the transcriptional activity of Smad, thereby assisting TGF-β/Smad dependent EMT [25]. The non-canonical pathways are systemically shown in Figure 2.

Phosphorylation of TGF-β type I and II receptors can also activate downstream non-canonical pathway including Rho, PI3K/Akt, and Grb2/SOS signaling in a Smad-independent manner.

## 3. TGF-β Signaling in Tissue Fibrosis

Tissue fibrosis is the major pathological feature of most chronic inflammatory diseases, which results in the failure of important organs and can lead to mortality [26]. In chronic kidney disease (CKD), fibrosis is an essential step in the development of end-stage renal disease (ESRD) [27,28,29,30,31], where TGF-β/Smad-dependent signaling pathway plays a critical role [4,32]. TGF-β1 medicates ECM synthesis and degradation in progressive renal fibrosis. TGF-β1 also induces the transformation of tubular epithelial cells to myofibroblasts through EMT to cause renal fibrosis [32,33]. The primary downstream mediators of TGF-β1, Smad2, and Smad3, are extensively activated in fibrotic kidneys in patients and animal models with CKD [34]. As fibrosis progresses, Smad2 is protective, while Smad3 leads to pathogenic changes. Smad3 directly binds to the promoter region of collagens to trigger renal fibrosis production and reduces the activity of MMP-1 to inhibit ECM degradation via induction of TIMP-1 [35,36,37,38]. By contrast, conditional knockout of Smad2 from tubular epithelial cells enhances Smad3-mediated renal fibrosis, which is associated with phosphorylation and nuclear translocation of Smad3, auto-induction of TGF-β1 expression, and transcription of collagen I and III genes [39]. In response to TGF- β1 and BMPs, Smad4 and Co-Smad promote nuclear translocation of Smad2/3 and Smad1/5/8 complexes, respectively [34,40]. Deficiency of Smad4 in mesangial cells in vitro dramatically suppresses collagen I promoter activity and significantly reduces fibrosis in the mice tubular epithelial cells with UUO-induced fibrosis without affecting Smad3 activation [41]. Smad7 is the inhibitor of TGF/Smad canonical signaling in fibrosis. Smad7 can compete with Smad2 and Smad3 for TGF-βR1 activation as well as ubiquitinated and degraded Smad2 and TGF-βRI by recruiting the E3 ubiquitin ligase Smad ubiquitination regulatory factors [42,43,44]. 

Extensive studies demonstrated the crucial role of TGF-β1 signaling in the pathogenesis of liver diseases, including hepatitis and cirrhosis as well as hepatocellular carcinoma [45]. In hepatic fibrosis, type I collagen expression and epithelial-myofibroblast transition are actively produced by up-regulating the pro-fibrotic SMAD3 but down-regulating the anti-fibrotic SMAD2. The responsive promoter activity of SMAD3 can be enhanced by SMAD4 but blocked by its native negative mediator SMAD7 [46]. 

Extensive studies demonstrated the crucial role of TGF-β1 signaling in the pathogenesis of liver diseases, including hepatitis and cirrhosis as well as hepatocellular carcinoma [45]. In hepatic fibrosis, type I collagen expression and epithelial-myofibroblast transition are actively produced by up-regulating the pro-fibrotic SMAD3 but down-regulating the anti-fibrotic SMAD2. The responsive promoter activity of SMAD3 can be enhanced by SMAD4 but blocked by its native negative mediator SMAD7 [46]. 

Heart failure is an increasingly prevalent disease in humans, a progressive loss of cardiomyocytes, ventricular chamber remodeling, and accumulation of interstitial fibrosis resulting in a lethal reduction of cardiac output [47]. Several studies have explored the potential contributions of both canonical and non-canonical pathways of the TGF-β signaling in the pathogenesis of cardiac fibrosis [48,49,50]. In addition, idiopathic pulmonary fibrosis (IPF) and interstitial lung fibrosis are particularly austere lung diseases [51], where TGF-β signaling is one of the most potent profibrotic inducer for accelerating the progression of lung fibrosis through recruiting and activating monocytes and fibroblasts as well as induction of ECM production in the lesion [52] (Figure 3). Importantly, increasing evidence revealed the importance of crosstalk between TGF-β and other signaling pathways in the pathogenesis of tissue fibrosis. For example, TGF-β1 increases the MMP production of lung stromal fibroblast through Wnt/β-catenin signaling pathway to future enhance the development of IFP [53]. Moreover, TGF-β signaling also works with a Hippo-YAP/TAZ pathway for modulating cardio fibrosis via up-regulating the production of TGF-β1 [54]. 

TGF-β signaling is a critical factor for initiating fibrogenesis in kidney, liver, heart, and lung. Phosphorylation of pro-fibrotic SMAD3 and SMAD4, and repression of anti-fibrotic SMAD2 and SMAD7 largely enhance the fibroblast proliferation, myofibroblast differentiation, and ECM production in the injured tissues. Accumulation of tissue stiffness and scar tissue largely affects their physiological functions and lead to organ failure.

## 4. TGF-β Signaling in the Tumor Microenvironment

Increasing evidence demonstrated an equal weight of both the adaptive and innate immunity in the cancer progression, which can be largely suppressed by cancer cells via TGF-β1 signaling. In adaptive immunity, elevated TGF-β-levels suppress T cell development and anticancer function [55]. TGF-β blocks Th1 and CD8+ cytotoxic T cells differentiation from naïve T cell by inhibiting the expression of T-box transcription factor (T-bet) at a transcriptional level [56,57]. TGF-β also supresses T cells proliferation via diminishing the expression of Interleukin-2 (IL-2) by promoting the interaction of Smad4 on an anti-proliferative mediator TOB at a genomic level [58]. In fact, early detection of SMAD4 in pancreas help stratify pancreatic cancer patients for new therapy selection [59]. Importantly, the regulatory role of TGF-β1 signaling has been emergingly observed in the innate immunity. Tumor-associated macrophage (TAM) can be further polarised to show pro-tumoral M2 phenotypes in the TME for exhibiting pro-angiogenic, anti-inflammatory, and immunosuppressive features to promote tumor development [60]. Recent evidence suggested that TGF-β can trigger the M1/M2 polarization of TAMs via activating the Smad2/3 and PI3K/AKT pathways for enhancing the transcription of pro-tumoral effectors, including IL-10, VEGFA and CXCR4 [61]. Moreover, TGF-β1/Smad3 signaling diminish the production of an anti-tumor cytokine, interferon-γ (IFN-γ) in natural killer cells (NK cells) by T-bet suppression to inhibit their cytotoxic function [62]. In addition, TGF-β1/Smad3 signaling can also markedly suppress the development of NK cells via downregulating the transcription factor E4BP4 via transcriptional regulation in a T-bet independent manner [9].

TGF-β is also crucial for activating cancer-associated fibroblasts (CAFs) to modulate migration and invasion of the cancer cells [63,64]. TGF-β is able to induce the transition process of fibroblasts towards myofibroblasts by increasing fibroblast contractility and CAF marker expression including α-smooth muscle actin (α-SMA), fibroblast activation protein (FAP), tenascin-C and platelet-derived growth factor receptor (PDGFR) [65,66,67,68]. Increasing evidence suggests that TGF-β1 can enhance the expression of ECM remodelling gene ACTA2 (α-SMA) [69,70], collagen precursor PLOD2, and membrane bound thymocyte differentiation antigen Thy1/CD90 for supporting the CAF formation [71]. Furthermore, angiogenesis is one of the critical steps in tumorigenesis, where TGF-β/Smads signaling is significantly involved in the regulatory machinery, such as vascular endothelial growth factor (VEGF), fibroblast growth factor-1 (FGF-1), and platelet-derived growth factor (PDGF) [72,73]. TGF-β also interacts with the type I receptors, activin receptor–like kinases 1 (ALK1) or 5 (ALK5) to regulate angiogenesis, where TGF-β/ALK1 signaling induces downstream signal via Smad1/5 while TGF-β/ALK5 uses Smad2/3 to regulate the angiogenic factors, respectively [73,74]. Activation of ALK1 requires activation of ALK5, as the lack of TGF-β/ALK5 activation will deteriorate both pathways, highlighting TGF-β as the key regulator to balance ALK1 and ALK5 in endothelial cells for mediating angiogenesis in cancer [73]. Indeed, TGF-β/Smad pathway also mediates endothelial-mesenchymal transformation via SNAIL/Slug expression in endothelial cells (TECs) to support sprouting angiogenesis, and accumulation of myofibroblast and CAFs in the TME [10] (Figure 4).

## 5. Importance of TGF-β Signaling in the Fibrotic TME

Fibrotic TME actively contributes to the pro-tumoral activities for promoting angiogenesis, ECM remodelling, immunosuppression, inflammation, and tumor growth [13,75]. CAFs are highly present in TME of most cancers associated with worse prognosis [76,77,78,79]. In fact, CAFs are highly heterogeneous in terms of their origin, phenotype, function, and location within the TME. CAFs originate from diverse cell types including resident tissue fibroblasts (via activation), fibrocytes (via recruitment), bone-marrow derived mesenchymal cells (via differentiation), epithelial cells (via EMT), endothelial cells (via endothelial-mesenchymal transformation, EndMT), stellate cells (via activation), pericytes, and adipocytes (via transdifferentiation). It is important to note that cancer cell-derived TGF-β1 have recently been implicated in the promotion of CAF heterogeneity [80].

### 5.1 Invasion and Metastasis

CAFs mediate migration and invasion of the cancer cells via altering the architecture of TME by various actions, including secretion of ECM (fibronectin and collagen type I, II), growth factors and cytokines (such as TGF-β, HGF, FGF, CXCL12, CCL2, CCL5, CCL7, CXCL16, and IL-6) [81,82,83], generation of escaping paths (via MMP, RhoA, ROCK, and MyoII) [84], and modification of the tumor ECM architecture and fibre alignment (via FAP, Cav1, SNAIL1, TWIST1, palladin, and mechanical force) [85,86,87]. TGF-β1 plays an important role in cancer migration due to its medication towards CAF contractility and MMP secretion, where CAFs produces MMPs to lose the structure of TME architecture via both canonical and non-canonical TGF-β signaling pathways, i.e., non-canonical pathways cause RhoA/ROCK to promote MyoII contractility, while canonical pathways enhance SNAIL1 and TWIST1 gene transcription for increasing CAF contractility [84,88]. In addition, Myosin II, α5β1 integrin and PDGFR α expression can also increase CAF contractility and traction force [89]. Similarly, tumor ECM stiffness can simultaneously activate the Rho switch in CAF and vice versa, in a feed-forward self-reinforcing loop to exacerbate CAF-mediated migration of cancer cells [90]. Furthermore, CAF-produced fibronectin displaying anisotropic orientation mediates the association between CAFs and cancer cells for permitting directional migration of the cancer cells [89]. Interestingly, TGF-β modulates the expression of fibronectin in CAFs, while fibronectin serves as a depot for acute activating the TGF-β latent complex [91,92]. TGF-β plays an essential role in the increment of CAF-mediated cross-linked collagen, which contributes to the formation of escaping paths for cancer cells [93]. TGF-β stimulates collagen production and collagen linearisation in CAFs together with the cancer cell secreted factors such as WNT1 inducible signaling pathway gene (WISP-1) and regulates LOX transcription via PI3K/Akt and MAPK signaling pathways [94,95]. Moreover, high concentration of collagen can increase the invasive capacity of the cancer cells in the TGF-β-rich TME [96].

### 5.2. Angiogenesis

Angiogenesis is important to ensure a continuous supply of essential nutrients and oxygen for cancer cells via expression of various pro-angiogenic factors such as vascular endothelial growth factor (VEGF), fibroblast growth factor 2 (FGF-2), fibroblast activation protein (FAP), and chemokine 12 (CXC12), which are maintained by CAFs via a secretome dependent manner [97]. Interestingly, CAF-produced CXC12 recruits endothelial progenitor cell (EPC) [98], where EPCs participate in neovascularization by differentiating into tumor-associated vascular endothelial cells as well as promoting and sustaining angiogenesis via paracrine mechanisms [99]. TGF-β is important in the regulation of CAF-mediated angiogenesis. TGF-β increases the expression of angiogenic effectors VEGF, CXC12, FGF, and MMP, which regulate ECM degradation and remodelling for facilitating angiogenesis. [100]. Furthermore, TGF-β/SMAD signaling promotes the expression of ECM proteins in fibroblasts via a TGF-β/fibronectin dependent axis, where fibronectin mediates vascular development and differentiation by facilitating the formation of perivascular matrix and mediating pericyte-endothelium association [101]. Moreover, TGF-β also inhibits angiopoietin-1, which is essential for vessel integrity maintenance in fibroblasts that increase the blood vessel permeability of the TME [102]. Moreover, TGF-β directly contributes to vascular development of TECs via activating ALK1 and ALK5 [103]. Smad2/3 activation triggers the ALK5 pathway to promote TECs aggregation and supports vessel maturation, while Smad1/5 regulates the ALK1 signaling pathway mediated ECs proliferation and migration for enhancing vessel formation [104]. 

### 5.3. Immunosuppression

It is well documented that stromal derived TGF-β1 can directly and indirectly modulate host immunity to achieve immunosuppression and immunosurveillance escape. TGF-β/Smad3 pathway can reduce the expression of class II major histocompatibility complex antigens to diminish the surface immunogenicity of cancer cells [105]. TGF-β1 can actively induce programmed cell death of B and T lymphocytes by upregulating lipid phosphatase (SHIP) and consequent inhibition of the PI3kinase/Akt survival pathway in a Smad-dependent fashion [106]. TGF-β1 blocks the development of CD8+ cytotoxic T lymphocytes by inhibiting c-Myc expression via a Smad/FOXP1 axis [58,107,108]. CD8+ activity is also supressed by CD4+CD25+ regulatory T cells in a TGF-β1 dependent manner [109]. Moreover, TGF-β1 promotes survival of post-thymic naïve CD4+ T lymphocytes to later promote CD4+CD25+ Treg differentiation from uncommitted peripheral CD4+ cells via FOXP3 and Smad3, while preventing its differentiation into the Th-1 and Th-2 effector cell, thereby leading to tumor cell tolerance [109,110,111]. In contrast, regulatory T cells (Tregs) secrete immunosuppressive TGF-β and IL10 to further amplify immunosuppressive effects of TGF-β [112]. 

It is well documented that fibrotic TME contributes to the immunosuppression of anticancer immunity via its constituents CAFs, fibrotic ECM, and fibrotic-induced hypoxia [113]. Among them, CAFs are the most important mediators to achieve generalised immunosuppression by reducing anti-tumor cytotoxicity, immunosurveillance escape and promoting tumor cell tolerance. CAF-secreted TGF-β hinders the maturation of dendritic cells by blocking the expression of co-stimulatory proteins (CD1a, HLA-DR, CD80, and CD86) that are essential for appropriate antigen presentation to reactive CD4+ lymphocytes [114]. Moreover, CAF-educated DCs secrete large amounts of immunosuppressive cytokine such as TGF-β, IL10, and mediate CD4+ t cell differentiation into Treg under TGF-β influence [115]. CAFs help circulating monocytes to differentiate into immunosuppressive M2-like phenotypes via a CXCL12-dependent mechanism [116]. Upon activation, M2-like TAMs produce angiogenic effectors (VEGF and PDGF), ECM components, growth factors, and cytokines (TGF-β, IL3, IL4, and IL10) [117]. Additionally, TAM-secreted TGF-β1 has been implicated in EMT induction via upregulation of SOX9 expression via both of Smad3- and C-Jun dependent pathways [118]. Moreover, TGF-β is important for obstructing macrophage differentiation towards M1-like phenotype as well as reducing its cytotoxic effect by inducing SNAIL and IRAK expression in the TME [61,119]. 

Indeed, CAFs also interfere with the cancer-killing capacity of NK cells by hindering their expression of cytotoxic molecules and cytokines via PGE2 and ISO, leading to impaired cytotoxic activation and expression of natural cytotoxicity receptors such as NKp44 and NKp30 [120,121]. TGF-β secreted from myofibroblast precursors not only inhibits proliferation and differentiation but also trigger apoptosis of the NK cells [122]. CAFs can reprogram immature myeloid cells into myeloid-derived suppressor cells (MDSCs), showing immunosuppressive phenotypes via stromal signaling [123]. CAF mediates the recruitment of both monocytic and granulocytic MDSC. The PDGFRα+ CAF produces granulocytic chemoattractants such as CXCL1 for recruitment of granulocytic MDSC, whereas FAP-expressing CAF mediates the recruitment and function of monocytic MDSC in a melanoma mouse model [124]. CAFs also produce monocytic chemoattractant CCL2 and M-CSF in the collagen-rich fibrotic TME [125,126,127]. Granulocytic MDSC supresses T cell proliferation, while monocytic MDSC inhibits T cell function and prevents its accumulation. MDSCs also respond to CAFs by influencing fibroblast activation [123]. The immunosuppressive effects of CAFs in the TGF-β1-mediated fibrotic TME are listed in Table 1.

### 5.4. Drug Resistance

Fibrotic TME is important in the development of primary and secondary drug resistance via various mechanisms [13,75], but the underlying mechanism is still largely unclear. TGF-β1 contributes to oxaliplatin resistance in human colorectal cancer via EMT [128]. TGF-β causes resistance of docetaxel in prostate cancer via inducing antiapoptotic gene Bcl-2 by acetylation of KLF5 [129]. Moreover, TGF-β1 can attenuate the tumor response to PD-L1 blockade by restricting the accumulation of CD8+ T cells in the TME [130]. Studies have shown that CAFs promote drug resistance by EMT regulation in fibrotic TME. CAFs can provide pyruvate and lactate to support their glycolytic metabolism in cancer cells, which ultimately induce drug resistance via TGF-β1-driven EMT for mediating apoptosis and drug influx restriction [131,132]. For example, CAFs significantly enhanced TGF-β1-driven EMT in lung cancer cell by IL-6 secretion, thereby reducing the sensitivity of cisplatin in NSCLC [133]. The IL-6 expression can also induce by a sub-phenotype of CAF, iCAF, which differentiated under TGF-β-mediated IL-1 suppression. iCAF then maintains tumor organoid proliferation by induction of the Janus kinase/signal (JAK-STAT) signaling pathway [80]. CAFs are required to switch to glycolytic metabolism by causing cytoplasmic translocation of G-protein-coupled oestrogen receptor (GPER) via a GPER/cAMP/PKA/CREB signaling pathway [134]. CAFs can also mediate chemoresistance to PD-1 immune inhibitor by myelocyte recruitment via FAP/CCL2 [135]. Furthermore, TGF-β2 and hypoxia-inducible factor (HIF1) secreted by CAF can also activate Hedgehog signaling pathway mediator GLI2 to promote cancer cell stemness and chemotherapy resistance in colorectal cancer, consequently [136]. 

## 6. Future Prospect

TGF-β signaling pathway is essential for several normal physiological processes, which largely hinders its translational development due to the potential side effects. For example, genetic deletion of TGF-β1 is associated with autoimmunity, abundant T-cell proliferation, activation, and Th1 differentiation [76]. In addition, disruption of TGF-β1/Smad3 signaling Smad3 in mice leads to impaired immunity, implying its vital importance in T cell development [137,138]. In recent clinical trials, None of the TGF-β inhibitors have been approved for cancer or fibrosis therapy due to its cell cytotoxicity. For example, TGF-β blocker is shown to induce bleeding and cardiac toxicity in several in vivo models [139]. Therefore, the choice of TGF-β downstream therapeutic target is then a fundamental aspect to address. Fortunately, several novel TGF-β1/Smad3-dependent downstream regulators have been discovered and showed therapeutic potential for tissue fibrosis without affecting the physiological functions of TGF-β1/Smad3 signaling (Table 2). Proto-oncogene tyrosine protein kinase Src and neural transcription factor Pou4f1 have been identified as key downstream regulators in the TGF-β1/Smad3 signaling for promoting renal fibrosis via macrophage-myofibroblast transition (MMT) [140,141]. ZEB1 is suggested to be upregulated and involved in the TGF-β-induced EMT, which can support metastasis and invasion of tumor cells by directly suppressing transcriptions of antifibrotic miR200c and miR141, where miR141 is an inhibitor of TGF-β as a feedback mechanism [142,143]. In addition, a novel Smad3-dependent long non-coding RNA Erbb4-IR showed various actions in prostate and esophageal squamous cell carcinoma, where one of the actions is to inhibit apoptosis by downregulating tumor suppressor miRNA-145, thus promoting cancer cell proliferation [68,144]. Upregulation of TGF-β/Smad-mediated Erbb4-IR and LRNA9884 were observed in renal fibrosis, as they up-regulate the pathogenic effectors via transcriptional regulation at genomic level [145,146,147,148]. The pathogenic roles of inflammatory cells in the TGF-β-mediated tissue fibrosis have been intensively elucidated especially on the innate immunity, e.g., macrophage, neutrophil, etc. [32,140,141,149,150,151]. Nevertheless, their potential functions in the TME with TGF-β pathway activation is still largely unclear and is starting to be recognized [12,152,153]. Therefore, better understanding of TGF-β downstream signaling would uncover precise therapeutic targets from the fibrotic microenvironment for cancer. Indeed, the potential implications of these newly discovered downstream regulators from the TGF-β1 signaling in the fibrotic TME are still largely unknown, and further investigation may identify their pathogenic roles in the fibrotic microenvironment as well as the therapeutic potentials in cancer.

## Figures and Tables

**Figure 1 ijms-22-07575-f001:**
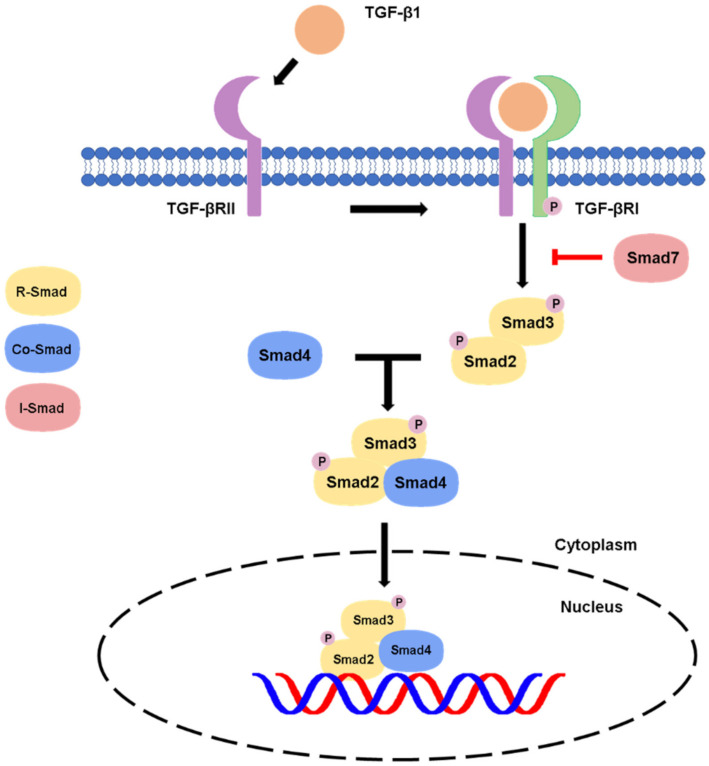
Canonical pathway of TGF-β signaling.

**Figure 2 ijms-22-07575-f002:**
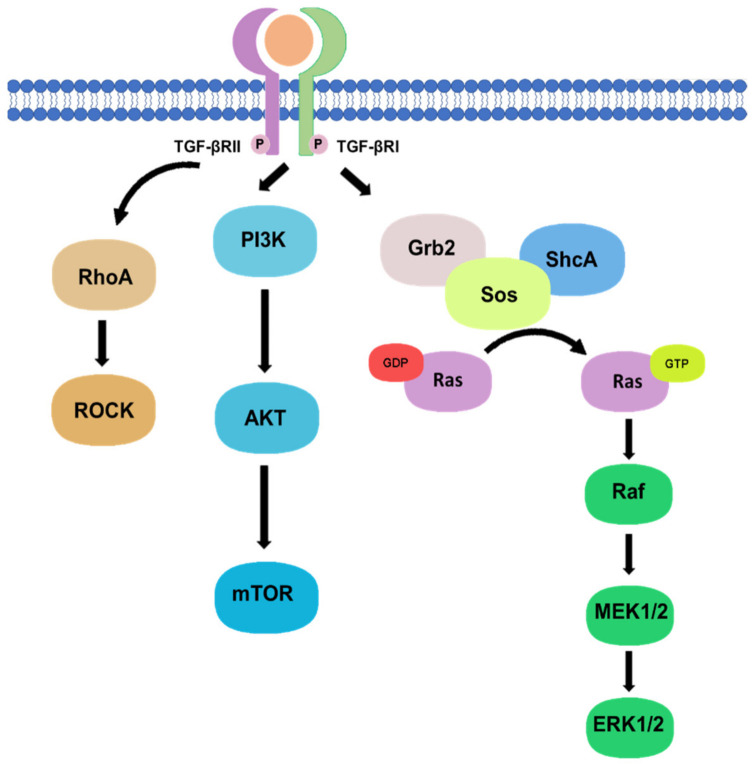
Non-canonical pathway of TGF-β signaling.

**Figure 3 ijms-22-07575-f003:**
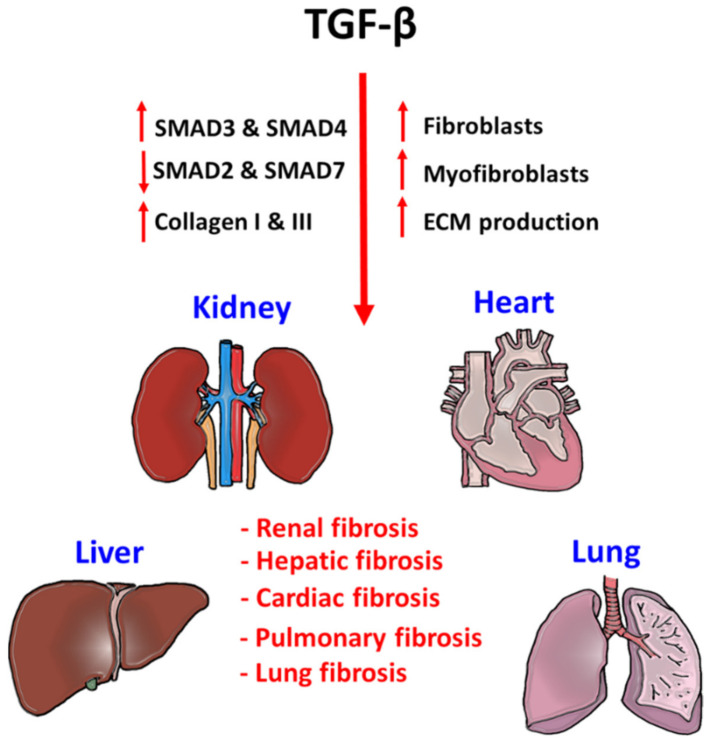
Pathogenic role of TGF-β signaling in tissue fibrosis.

**Figure 4 ijms-22-07575-f004:**
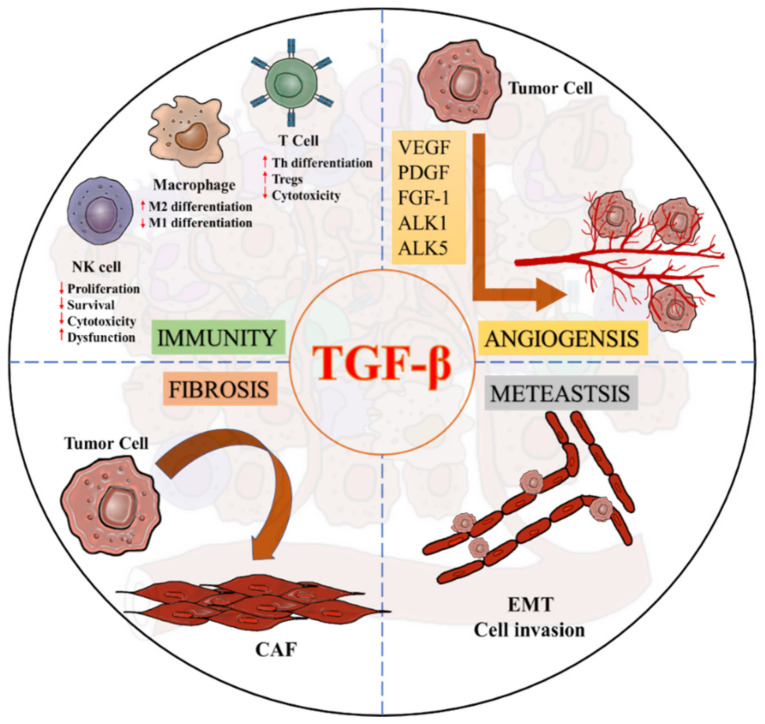
Diverse roles of TGF-β signaling in a pro-tumoral TME.

**Table 1 ijms-22-07575-t001:** Immunosuppressive effects of CAFs in the TGF-β1-mediated fibrotic TME.

Cell Type	Effects of CAF on Immune Cell	Ref.
Dendritic cell	Inhibiting DC maturation	[114]
Macrophage	Promotes macrophage differentiation into immunosuppressive M2-like phenotype and inhibiting differentiation into pro-inflammatory M1-like phenotype.	[61,116]
Natural killer cell	Impairs cytotoxic activation and the expression of natural cytotoxicity receptors, as well as promoting NK cells apoptosis	[120,121]
T cell	Reduces T-cell activation, differentiation by influencing DC and MDSC	[115,124]
MDSC -granulocytic -monocytic	Reprograms myeloid cells into MDSC Increases its recruitment and function to supress T cell proliferation Increases its recruitment to inhibit T cell activity and accumulation in TME	[123,124]

**Table 2 ijms-22-07575-t002:** The downstream therapeutic targets of TGF-β signaling in fibrotic microenvironment.

Factor	Mediators	Effect After Inhibition	Ref.
MicroRNAs (miRNAs)	miR-21	Decreases ECM synthesis and fibrosis via upregulation of MMP-9 expression	[154]
	miR-200	Protects TGF-β-mediated EMT by inhibiting the expression of ZEB 1 and 2	[155]
Long non-coding RNAs (LncRNAs)	Lnc-LFAR1	Downregulates TGF-β/Smad signaling pathway cancer cell proliferation	[156]
		Blocks tissue fibrosis by reducing Smad2/3 phosphorylation and binding to TGFβR1	[157]
	Lnc-TSI	Upregulates the interaction between Smad3 and TGFβR1 lead to cancer metastasis	[158]
	LncRNA H19X	Reduces ECM synthesis induced by TGF-β and controls the differentiation and survival of ECM-producing myofibroblasts	[159]
	Erbb4-IR	Supresses miR-29b transcription and consequent antifibrotic function	[68]
		Downregulates miRNA-145 to reduce cancer cell proliferation	[144]
Transcription factors (TFs)	Twist	Increases sensitivity to chemotherapy by downregulating MDR1 and reducing drug efflux	[160]
	ZEB1	Reduces antifibrotic miR200c and miR141 expression	[142]
		Reverses metastasis and restores chemosensitivity in chronic chemoresistant	[161]
		Reduces chemoresistance mediated by MGMT	[162]
		Restores miR-203 expression, supresses stemness and promotes chemo-sensitivity	[163]
	Pou4f1	Prevents macrophage-myofibroblast transition, thereby supressing MMT-mediated fibrosis	[140,164]

## Data Availability

Not applicable.

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
