# Peer review of "TGF-β Signaling: From Tissue Fibrosis to Tumor Microenvironment"

_ijms, 2021, doi:10.3390/ijms22147575_

Round 1
Reviewer 1 Report
The review is well written and describes key concepts of TGF-β Signaling. However, I have following critiques :
1.The review should have more figures. The key concepts especially signaling eg Non-canonical pathway, TGF-β signaling in tumor microenvironment, TGF-β signaling in tissue fibrosis etc. should also be explained by the figures. This will make the review more interesting to the readers.
2. The authors should make headings and sub-headings more differentiated by changing the font size etc.
3. The authors should also elaborate more on "future prospects' section.
Author Response
1.The review should have more figures. The key concepts especially signaling eg Non-canonical pathway, TGF-β signaling in tumor microenvironment, TGF-β signaling in tissue fibrosis etc. should also be explained by the figures. This will make the review more interesting to the readers.
Response: Thanks for your valuable comments. The mentioned key concepts are further explained with new Figures 2, 3 and 4.
2. The authors should make headings and sub-headings more differentiated by changing the font size etc.
Response: Thanks for the suggestion. Font sizes of the headings and sub-headings are changed to make them more distinguishing. Thank you.
3. The authors should also elaborate more on "future prospects' section.
Response: This section is further enriched with additional information about some newly identified therapeutic targets from the downstream of TGF-beta signaling.
Reviewer 2 Report
I will be happy to re-evalute this manuscript after major English correction / proofreading. There are too much important issue in this manuscript to express an objective evaluation. This reviewer acknowledges the potential interest of this paper, but it should be drastically improved in the English language before an actual peer-review. Otherwise it should be rejected.
Author Response
Response: Thank you very much for your interest in this manuscript. To address your concern about the English, we have done a major correction and proofreading of the manuscript to improve its readability. Looking forward to your evaluation.
Round 2
Reviewer 1 Report
My critiques have been addressed.
Author Response
Thank you for reviewing our manuscript
Reviewer 2 Report
This is an interesting paper on the role of TGF beta in tissue biology and cancer, timely and clear. Also the figures are very helpful for the readers. This reviewer acknowledges also a good citations’potential. Here some suggestions to improve the paper:
- fibrosis: please expand the part on tissue fibrosis also including parallelism with other pathways, as for example Wnt pathway in pulmonary fibrosis;
- SMAD4 expression (IHC) and potential importance for therapy selection in pancreatic cancer (Iacobuzio-Donuhae, J Clin Oncol 2009 and others more recent)
- please comment on tumor microenvironment (inflammatory cells) and TGF beta pathway activation. Any peculiarities?
Thank you
Author Response
fibrosis: please expand the part on tissue fibrosis also including parallelism with other pathways, as for example Wnt pathway in pulmonary fibrosis;
Response: Thanks for the valuable comments. The mentioned pathway such as Wnt and Hippo-Yap are further explained in tissue fibrosis part.
2. SMAD4 expression (IHC) and potential importance for therapy selection in pancreatic cancer (Iacobuzio-Donuhae, J Clin Oncol 2009 and others more recent)
Response: Thanks for the suggestion. The potential importance for SMAD4 is now include in the tumor microenvironment section. Thank you.
3. please comment on tumor microenvironment (inflammatory cells) and TGF beta pathway activation. Any peculiarities?
Response: The pathogenic roles of inflammatory cells and TGF beta pathway are further comment in future prospect section.